# High Performance Low-Temperature Solid Oxide Fuel Cells Based on Nanostructured Ceria-Based Electrolyte

**DOI:** 10.3390/nano11092231

**Published:** 2021-08-29

**Authors:** Jiamei Liu, Chengjun Zhu, Decai Zhu, Xin Jia, Yingbo Zhang, Jie Yu, Xinfang Li, Min Yang

**Affiliations:** Key Laboratory of Semiconductor Photovoltaic Technology of Inner Mongolia Autonomous Region, School of Physical Science and Technology, Inner Mongolia University, 235 West Daxue Street, Hohhot 010021, China

**Keywords:** LT-SOFCs, doped ceria, ceria-based electrolyte, electrochemical properties

## Abstract

Ceria based electrolyte materials have shown potential application in low temperature solid oxide fuel cells (LT-SOFCs). In this paper, Sm^3+^ and Nd^3+^ co-doped CeO_2_ (SNDC) and pure CeO_2_ are synthesized via glycine-nitrate process (GNP) and the electro-chemical properties of the nanocrystalline structure electrolyte are investigated using complementary techniques. The result shows that Sm^3+^ and Nd^3+^ have been successfully doped into CeO_2_ lattice, and has the same cubic fluorite structure before, and after, doping. Sm^3+^ and Nd^3+^ co-doped causes the lattice distortion of CeO_2_ and generates more oxygen vacancies, which results in high ionic conductivity. The fuel cells with the nanocrystalline structure SNDC and CeO_2_ electrolytes have exhibited excellent electrochemical performances. At 450, 500 and 550 °C, the fuel cell for SNDC can achieve an extraordinary peak power densities of 406.25, 634.38, and 1070.31 mW·cm^−2^, which is, on average, about 1.26 times higher than those (309.38, 562.50 and 804.69 mW·cm^−2^) for pure CeO_2_ electrolyte. The outstanding performance of SNDC cell is closely related to the high ionic conductivity of SNDC electrolyte. Moreover, the encouraging findings suggest that the SNDC can be as potential candidate in LT-SOFCs application.

## 1. Introduction

Solid oxide fuel cells (SOFCs), as a kind of clean energy device which can directly convert chemical energy into electric energy, have attracted wide concern [1,2,3]. Generally, conventional SOFCs require high working temperature over 800 °C, which not only narrows the selection range of electrolyte materials, but also limits the practical applications of cells. Therefore, it is necessary to lower the traditional operating temperature to the range of low temperature (400–600 °C). Currently, a great deal of effort has been focused on the research of LT-SOFCs. One of the key challenges in developing LT-SOFCs is to seek electrolyte materials with high ionic conductivity as alternatives of traditional yttrium-stabilized zirconium (YSZ) electrolyte. Extensive attempts have been devoted to obtaining highly ionic conductors by exploring the new electrolyte materials or optimizing the existing ones.

In recent years, cerium-based oxide (CeO_2_) has attracted extensive interests as the electrolyte in the fields of fuel cell due to the characteristic to store and release oxygen via facile Ce^4+^/Ce^3+^ redox cycles [4,5,6]. However, pure CeO_2_ has poor thermal stability and can be easily sintered at high temperature, which causes a rapid decrease in its oxygen storage/release capacity and catalytic activity [7,8]. In addition, pure CeO_2_ itself is an insulator, its ionic conductivity is very low (10^−5^ S·cm^−1^) [9]. One approach to overcome this limitation is to use the doping with lower valence metal cations, especially for some trivalent rare earth (RE) cations. The doping of RE cations can lead to the ceria lattice disorder and create stress in the ceria lattice, which could increase oxygen mobility from ceria lattice to its surface, and decrease the activation energy of oxygen vacancy formation because of the charge neutrality and nonstoichiometry compensation [10]. Therefore, it is expected that the oxygen ionic conductivity can dramatic rise due to the increase in the oxygen vacancy concentration causing by the RE doping. For example, the conductivity of Sm-doped ceria (Ce_0.8_Sm_0.2_O_1.9_, SDC) is 0.0114 S·cm^−1^ at 600 °C [11], while Gd-doped ceria (Ce_0.9_Gd_0.1_O_1.95_, GDC) obtains a conductivity of 0.019 S·cm^−1^ at 600 °C [12]. The Pr-doped CeO_2_ can improve the ionic conductivity by 1–3 orders of magnitude [13]. It was found that oxygen vacancies can also be promoted in the ceria lattice by other aliovalent doping of lower valence cations Nd^3+^, Ga^3+^ or Ca^2+^ [14]. Another promising approach is co-doped or multiple ion doping route, i.e., ceria doped with two or more aliovalent cation, which has been found to show higher ionic conductivity than single coped ceria. Banerjee et.al. [15] reported that Ca and Sm co-doped ceria displayed an amazing ionic conductivity of 0.122 S·cm^−1^ at 700 °C. Sr and Tm co-doped ceria designed by Zhu produced the ionic conductivity of 0.13 S·cm^−1^ at 550 °C [16]. Fan et.al. [13] developed the SDC with Nd^3+^ and Pr^3+^ dopants and the ionic conductivity of the doped ceria reached 0.125 S·cm^−1^ at 600 °C. Aliye Arabacı et al. [17] studied the performance of Gd^3+^ and Nd^3+^ co-doped CeO_2_ electrolyte (GNDC) materials, and found that the GNDC ionic conductivity was significantly improved compared with undoped ceria. Shobit Omar et al. [18,19] found the ionic conductivity of Sm^3+^ and Nd^3+^ co-doped ceria (SNDC) to be 30% higher that of GDC at 550 °C. These results demonstrate that double or co-doping is a successful strategy and can greatly promote the ionic conductivity of ceria-based electrolytes.

In addition, the nanotechnology approach is also very effective in enhancing the ionic conductivity of ceria-based oxides. And it has aroused extensive interests of scientist community in gaining good ionic conductivity. Recently, nanocrystalline ceria alone have been introduced as the electrolyte for advance LT-SOFCs. Using nanocrystalline electrolytes, the ionic conductivity can be further improved. Takamura [20] prepared the CeO_2_-based nanoparticles and acquired the conductivity of 0.003 S·cm^−1^ at the low temperature (300 °C). Chen et al. [21] reported that the nanocrystalline GDC electrolyte generated a remarkable power out of 591.8 mWcm^−2^ with extraordinary ionic conductivity of 0.37 S·cm^−1^ at 550 °C. The out performance is more than 3 times higher than that of traditional high-temperature sintered GDC electrolyte. The nanocrystalline electrolyte has presented unexceptionable properties as compared to the conventional electrolyte, which revealed that the interface/surface conduction of nanocrystalline GDC electrolyte was critical in improving ionic conductivity.

Given that the advantages of co-doped strategy and nanocrystalline electrolyte, in this work, we synthesized a Sm_0.075_Nd_0.075_Ce_0.85_O_2-δ_ (SNDC) material with better performance than GDC via Sm^3+^ and Nd^3+^ co-doped ceria strategy, and further fabricated it into the nanocrystalline SNDC electrolyte. Various experimental characterizations were carried out to investigate the phase structure, microstructure, electrical and electrochemical properties, as well as fuel cell performances of the nanocrystalline SNDC and pure CeO_2_ electrolytes. The surface properties of the materials were studied to understand its conduction behavior. Remarkably, the Sm^3+^ and Nd^3+^ co-doped ceria exhibited high ionic conductivity and displayed excellent fuel cell performance of 1070.31 mW·cm^−2^ at 550 °C. Here the mechanisms on the greatly enhanced electrical property is also discussed.

## 2. Experimental

### 2.1. Fabrication of Materials

Sm^3+^ and Nd^3+^ co-doped CeO_2_ (Sm_0.075_Nd_0.075_Ce_0.885_O_2-δ,_ SNDC) and pure CeO_2_ powders were prepared by glycine-nitrate method. Stoichiometric amounts of Ce (NO_3_)_3_·6H_2_O (≥99%), Sm_2_O_3_ (99.9%), Nd_2_O_3_ (99.9%), C_2_H_5_NO_2_ (99.5~100.5%), HNO_3_ (65~68%) were used as original materials. The corresponding mass of Sm_2_O_3_ and Nd_2_O_3_ were weighed and dissolved completely in an appropriate amount of concentrated nitric acid to form nitrate solution, respectively. Further, Ce (NO_3_)_3_·6H_2_O was dissolved in distilled water. Then, they were mixed together to obtain a mixed solution, and an appropriate amount of glycine (C_2_H_5_NO_2_) was added to the solution at the ratio of glycine to metal cation was 1.2:1. The mixed solution was heated and stirred slowly in a constant temperature magnetic stirrer until it formed a glutinous colloid and finally boiled and spontaneously ignited, resulting in a fluffy yellowish powder. The precursor powders were calcined at 800 °C for 2 h to remove the residual organic matter in the sample, and the cubic fluorite structure SNDC powder sample was obtained. CeO_2_ powders were also prepared by the similar process.

### 2.2. Fabrication of Fuel Cells

The two types of fuel cells were prepared with the nanocrystalline structure SNDC and CeO_2_ electrolytes, respectively. Foam-Ni-NCAL was used as the electrode layer, and SNDC or CeO_2_ powders was used as electrolyte layer. The foam-Ni-NCAL electrode layer was prepared using commercial Ni_0.8_Co_0.15_Al_0.05_LiO_2-δ_ (NCAL) oxide powder (Tianjin Bamo Science and Technology Joint Stock Limited, Tianjin, China) and nickel foam (Ni-foam). The Ni-foam was first made into a round shape with a diameter of 13 mm using a punch. The NCAL powders, terpineol and alcohol were mixed into slurry, and the slurry was coated onto the Ni-foam and dried at 120 °C to form the electrode layer (foam-Ni-NCAL) of the cell. Then a piece of the foam-Ni-NCAL electrode was placed at the bottom of the mold, then 0.32 g SNDC electrolyte powder was weighed and spread evenly as the middle layer, and then another foam-Ni-NCAL electrode was placed on the SNDC powder. Finally, the three layers with electrodes and electrolyte were pressed under 450 MPa into a single cell constructed in a symmetrical configuration of foam-Ni-NCAL/SNDC/NCAL-Ni-foam. The effective electrode area of the cell was 0.64 cm^2^. The thickness and diameter of the cell are 2 mm and 13 mm, respectively. Hydrogen and air were passed into the electrode for measurement as fuel and oxidant, respectively. The foam-Ni-NCAL/CeO_2_/NCAL-Ni-foam cell with the nanocrystalline structure CeO_2_ electrolyte was also made by the same method, as described above.

### 2.3. Characterizations

The phase structures of CeO_2_ and SNDC samples were determined through a powder X-ray diffraction (XRD, MiniFlex 600, Rigaku Corporation, Rigaku, Japan) at room temperature and 2*θ* varying from 20° to 80° by steps of 10°. Raman spectra analysis was performed by using a Horiba Raman scope (Raman, Horiba Jobin Yvon U1000, HORIBA Scientific, Paris, France) equipped with an Olympus LMPlan optical microscope and a charge-coupled device (CCD) camera. The microstructural characterization was carried out using scanning electron microscopy (SEM, Phenom Pro, Phenom-World B.V., Eindhoven, Netherlands), High resolution transmission Electron Microscope (HR-TEM, Titan G2 60-300, FEI, Hillsboro, OR, USA) and Energy dispersive X-ray analysis (EDS, X-Max^N^150, Oxford Instruments, Oxford, UK). The oxidation states and surface chemical properties of CeO_2_ and SNDC samples were studied by X-ray photoelectron spectroscopy (XPS, ESCALAB-250Xi, Thermo-fisher Scientific Co., Thermo-fisher Scientific Co., Waltham, MA, USA). The current–voltage performances of the symmetrical cells were tested at temperatures from 450 to 550 °C using an electronic load (ITECH DC ELECTRONIC LOAD, IT8511, ITECH Electrical Co., Ltd., Nanjing, China), H_2_ and air were supplied as the fuel and the oxidant, respectively. The flow rates of both H_2_ and air were 120 mL/min. The electrochemical impedance spectra (EIS) measurements were determined on the symmetric cells under open-circuit conditions using Zahner Zennium electrochemical workstation (EIS, Thales Z2.29 USB, Zahner Zennium, Germany Zahner electrochemistry Company, Kronach, Germany). The frequency range of measured EIS of the cells was 0.1–10^6^ Hz with an AC amplitude of 5 mV.

## 3. Results and Discussion

### 3.1. Results of XRD

The phase structures of the CeO_2_ and SNDC materials sintered at 800 °C for 2 h were analyzed by X-ray diffraction experiments. The testing range of 2*θ* varied from 20° to 80°. As shown in Figure 1, the diffraction peaks located at 2*θ* = 28.5°, 33.1°, 47.4°, 56.3°, 59.1°, 69.4° and 76.6°, can be assigned to the (111), (200), (220), (311), (222), (400) and (331) indexed characteristic peaks with cubic fluorite structure of CeO_2_ according to PDF#43-1002 [13]. The Co-doped SNDC electrolyte materials still have the same phase structure as CeO_2_, both of which belong to cubic fluorite phase, indicating that Sm^3+^ and Nd^3+^ have successfully replaced the position of Ce^4+^ in the lattice and traces of impurity phase are not observed. Moreover, the peak position of SNDC is slightly shifted to lower angle (Figure 1b). According to the Bragg formula, the crystal lattice becomes larger after doping, which is mainly attributed to the lattice expansion caused by the bigger ionic radius of Sm^3+^ (1.079 Å) and Nd^3+^ (1.109 Å) substitutions at Ce^4+^ (0.97 Å) sites [11,22].

### 3.2. Raman Spectra Analysis

Raman spectroscopy is used to reveal deeply the presence of oxygen vacancies caused by the change in the structure of the lattice chemical expansion owing to producing stresses of constrictions by the dopant cation. Figure 2a shows the Raman spectra of the pure CeO_2_ and SNDC samples from 200–800 cm^−1^. The results of the Raman spectra are obtained and analyzed by using the excitation laser line of 514 nm. It is known that the CeO_2_ has a cubic fluorite structure, and belongs to the Fm-3 m space group. As depicted in Figure 2a, pure CeO_2_ has only a single allowed Raman mode and shows a strong band at about 464.2 cm^−1^ (Peak α), which has F_2g_ symmetry and can be assigned to a symmetric breathing mode of the oxygen atoms around Ce-O_8_ vibration unit. In comparison to pure CeO_2_, the Sm^3+^ and Nd^3+^ Co-doped sample (SNDC) has shown a small systematic shift of the F_2g_ mode towards lower frequency, and the F_2g_ peak becomes wider and its intensity also becomes weaker. As reported, the shift of the F_2g_ mode maybe result from the expansion of ceria lattice and the increase of oxygen vacancies, while the shape change in the Raman line is mainly related to the lattice distortion of CeO_2_ induced by the substitution of Sm^3+^ and Nd^3+^ ions [11,23,24,25]. In addition, in the Sm^3+^ and Nd^3+^ co-doped ceria sample, two additional weak second order peaks at about 553.9 cm^−1^ (Peak β) and 605.5 cm^−1^ (Peak γ) are detected (Figure 2b), which can be attributed to defect spaces owing to extrinsic oxygen vacancies or structural defects caused by Sm^3+^ and Nd^3+^ dopants in ceria lattice [26,27].

### 3.3. X-ray Photoelectron Spectroscopy (XPS) Analysis

XPS is employed to analyze the element valence states so as to study the surface chemical properties of CeO_2_ and SNDC samples. Figure 3a shows the XPS full spectra of samples, and it is clear that there are Ce, Sm, Nd, O, and C characteristics in SNDC samples. In order to determine the chemical state, the core-level spectra of O 1s, Sm 3d, Nd 3d and Ce 3d are discussed in detail, and the multi-component XPS correlation peaks are analyzed by Shirley-type background subtraction method. The three peaks of oxygen are related to the divergence of chemical environment in the sample, which are divided into lattice oxygen, defect oxygen and oxygen vacancy [28]. The O 1s XPS core-level spectra of the CeO_2_ and SNDC samples (Figure 3b) shows two types of different oxygen species features, including the chemically adsorbed oxygen species (O_ads_) and the lattice oxygen (O_latt_) [29]. In general, the peak at 527.2–529.4 eV are attributed to O_latt_ (O^2−^), while the signal of the binding energy at 529.5–532.1 eV corresponds to the oxide defects or the surface oxygen species (O_ads_) adsorbed on the oxygen vacancies (i.e., O^−^, OH^−^, and CO_3_
^2−^) [30]. Usually, once the doped Sm^3+^ and Nd^3+^ are incorporated into the ceria lattice, the related Ce−O−Sm or Ce−O−Nd bonds are formed, which makes the migration process from the inner lattice oxygen to the surface lattice oxygen easier because of a slight change in electro-negativity of the guest and host ions [11,31]. In Sm^3+^ and Nd^3+^ co-doped CeO_2_, it is interesting to observe that the O_ads_/O_latt_ value in the SNDC sample is 0.84, which is greater than the value of un-doped CeO_2_ (0.32). This indicates that the lattice oxygen in the double-doped sample is significantly reduced and the oxygen ion transfer number is increased. Figure 3c shows the Ce 3d core-level spectra with Ce 3d_5/2_ and Ce 3d_3/2_ states of SNDC and pure CeO_2_ samples. The spectra deconvolution reveals that the presences of the peaks mainly result from Ce 3d_5/2_ and Ce 3d_3/2_ contributions, and the “v” and “u” peaks correspond to Ce 3d_5/2_ and Ce 3d_3/2_ states, respectively [16,32,33]. Moreover, the peaks labelled with v, v″, v″′, and u, u″, u″′ are due to cerium ions in the 4+ state, while v_0_, v′, and u_0_, u′ are assigned to 3+ ions [27,34], which clearly indicate the coexistence of Ce^4+^ and Ce^3+^ ions on the surface of the samples. It has been reported that the existence of Ce^3+^ is generally associated with the formation of oxygen vacancies and a higher Ce^3+^ concentration implies larger amounts of oxygen vacancies [35]. As shown in Figure 3c, the sum of the spectral peak areas related to the Ce^3+^ increases in SNDC compared with CeO_2_, implying that higher Ce^3+^ ion concentration can be obtained in SNDC sample [36], which facilitates the redox cycles between Ce^3+^ and Ce^4+^, and results in more oxygen vacancies on the surface. Figure 3d,e shows the Sm3d and Nd 3d XPS spectra of the as-prepared SNDC sample. In the case of Sm 3d spectra (Figure 3d), two single peaks are fitted with a pair of Sm 3d_5/2_ (located at 1109.62 eV) and Sm 3d_3/2_ (located at 1082.08 eV) spin-orbit coupling components by only considering the spin-orbit splitting. The XPS results for the Nd 3d spectra (Figure 3e) fitted into the distinguishable peaks show that the oxidation states of Nd in the compound are 3+ and 4+ states, and Nd^3+^ and Nd^4+^ coexist in the sample. More importantly, it clearly shows that the spectra peak area of Nd^3+^ is obviously larger than that of Nd^4+^. This suggests that the oxidation state of Nd^3+^ is dominant in the SNDC sample. As reported, with the increase of temperature, the high valence cation will be reduced to low valence cation, resulting in the formation of more oxygen vacancies [37,38]. This is very beneficial to the migration of oxygen ions and the improvement in ionic conductivity and electrochemical performance of SOFCs.

### 3.4. EDX and HR-TEM Analysis

The EDX element analysis of pure CeO_2_ and SNDC samples is presented in Figure 4a–h. A similar distribution of elements can be observed for the pure CeO_2_ (Figure 4a–c) and SNDC (Figure 4d–h). This demonstrates the homogeneous Sm^3+^ and Nd^3+^ ions distribution in the CeO_2_ lattice. This result provides good agreement with the XRD results of SNDC sample. It also confirms that the Sm^3+^ and Nd^3+^ ions have been successfully doped into the CeO_2_ lattice through co-doped approach, instead of forming second phases. Figure 4i shows the HR-TEM images of nanocrystalline SNDC powder prepared by the glycine-nitrate method. As shown, the SNDC powders are consisted of nano-sized particles and have some agglomeration. The sample exhibits large domains of SNDC, with a certain ordered framework, and non-uniform nanoparticles with a crystallite size ranging from 20 to 40 nm, interconnected to each other. Figure 4j shows the HR-TEM images at larger magnification of the raw nanocrystalline SNDC powder. It shows that the SNDC is highly crystalline and display an interplanar spacing of 0.31 nm, corresponding to the (111) lattice planes of SNDC.

### 3.5. Fuel Cell Performance Analysis

Figure 5a,b shows I–V and I–P characteristics for single cells with nanocrystalline structure CeO_2_ and SNDC electrolytes and the corresponding simple schematic diagram of the cell. As reported previously, the open circuit voltages (OCVs) of cells increase rapidly at the beginning of the test and then stays above 1 V, which is closed to the value of SOFC reported in the literature [21], indicating that the nanoelectrolytes are effective at separating H_2_ and air from each side of the cell. The achieved OCVs of the cell for the nanocrystalline structure SNDC electrolyte (Figure 5a) reach the 1.154, 1.148 and 1.141 V at 450, 500 and 550 °C, respectively, which are more than those (1.14, 1.133 and 1.078 V at corresponding temperatures) of cell with the nanocrystalline structure CeO_2_ electrolyte (Figure 5b). The higher OCVs indicated that the nanocrystalline structure CeO_2_ and SNDC, as the electrolytes of cells, can prevent the ceria electronic current leakage and achieve better performance. Correspondingly, the peak power densities of reach 406.25, 634.38, and 1070.31 mW·cm^−2^ for the nanocrystalline SNDC, but those of the nanocrystalline CeO_2_ cell were 309.38, 562.50, and 804.69 mW·cm^−2^, respectively. The peak power densities are increased by an average of about 26% compared with those of CeO_2_ cell. The significant enhancement of the power density reflects higher ionic conductivity of the SNDC electrolyte and lower electrode polarization resistance.

In order to investigate the electrochemical mechanism in the cells, the SEM microstructure and EIS analysis were carried out. Figure 6 shows the cross-sectional SEM images of the two symmetrical cells with a foam Ni–NCAL/electrolyte/NCAL–Ni foam structure after the measurement. From the Figure 6, it can be observed the two cells have all similar microstructures. The symmetric foam Ni-NCAL electrodes show porous structure and closely combined with electrolytes (Figure 6b,e), while the nanocrystalline electrolytes have no obvious pores, and both pure CeO_2_ and SNDC electrolytes present approximately 650 μm thick (Figure 6a,d). From Figure 6c,f, the pure CeO_2_ and SNDC electrolytes display the nanocrystalline microstructures, and the nanocrystalline electrolyte particles are below 50 nm. However, the SNDC exhibits even smaller nanocrystalline particles. Therefore, the significant difference in electrochemical performance of the cells can be attributed to the change in ionic conductance caused by the lattice expansion after double-doping.

Figure 7a,b shows the impedance spectra of the cells with the nanocrystalline structure CeO_2_ and SNDC electrolytes under H_2_/air condition at 450–550 °C. The equivalent circuit obtained according to the impedance spectrum fitting is shown in the Figure 7c. The formula used for fitting is R_0_ (Q_1_R_1_) (Q_2_R_2_), where R_0_ is the overall ohmic resistance, R_1_ and R_2_ correspond to the high and low frequency resistance arcs from the electrode, respectively, and Q is a constant phase element. The high frequency (R_1_, Q_1_) arc is usually attributed to ion charge transfer process and the low frequency (R_2_, Q_2_) arc is commonly associated with oxygen dissociation and surface diffusion processes. The high-frequency intercept on the real axis represents the ohmic resistance (R_0_). The difference between the high- and low-frequency intercepts on the real axis is related to the electrode polarization resistance (R_p_). It can be seen from Figure 7 that at 550 °C, the R_0_ value for the CeO_2_ cell was 0.12 Ω·cm^2^, and that for SNDC cell was 0.11 Ω·cm^2^, while the R_p_ value for the CeO_2_ cell was 0.55 Ω·cm^2^, and that for SNDC cell was 0.44 Ω·cm^2^. Obviously, compared with CeO_2_ cell, SNDC cell has smaller R_0_ and R_p_ values. In particular, the R_p_ value of SNDC cell is much lower than the R_p_ value of the CeO_2_ cell. Since the two cells adopt a foam Ni–NCAL/electrolyte/NCAL–Ni foam symmetrical structure and both use foam Ni–NCAL as the electrode, the intrinsic property of these electrodes should be the same. Moreover, the NCAL is an electron conductor, and the NCAL combined with the nickel foam has also high conductivity. Therefore, the R_0_ mainly originates from the electrolytes, and the differences between their R_p_ are most likely owing to the difference in the nanocrystalline electrolytes used. Combined with the results of the Raman Spectra and XPS analysis, it can be seen that the difference in R_p_ of the two electrolytes can be attributed to the different electrochemical performance in the three-phase interface caused by Sm^3+^ and Nd^3+^ dopants in ceria lattice. After double doping, the active site of the interface between electrolyte and electrode increases, and the three-phase interface increases, which eventually leads to the decrease of the R_p_ of the cell. In addition, as for SNDC, the ionic activity increases with the increase of test temperature, so the R_p_ shows a decreasing trend. The lower R_0_ in SNDC indicates that the SNDC electrolyte has higher ionic conductivity than that of the pure CeO_2_. In ceria-based electrolyte, the O^2−^ transport relies on the charge transfer process during the redox process. The XPS analysis has shown that the Ce^3+^ ion concentration increases in SNDC. Therefore, at the surface of the SNDC particles, oxygen ion transfer can be facilitated by the electron exchange resulting from the redox between Ce^3+^/Ce^4+^. As reported, the oxygen reduction reaction (ORR) process was closely associated with the electrode polarization loss of LT-SOFCs [39]. In this study, the lithiated transition metal oxide (NCAL) was employed as symmetrical electrodes. The NCAL in SOFCs exhibits superior catalytic activity for ORR in cathode and hydrogen oxidation reaction (HOR) in anode [40]. The symmetrical NCAL electrodes can improve the anodic HOR process, while enhancing the cathodic ORR process, and can generate more active oxygen species (e.g., O^2−^), which accumulate at the triple-phase boundary. The higher O^2−^ ionic conductivity in SNDC electrolyte promotes the transport of the active oxygen species (O^2−^) through the SNDC electrolyte. This reduces the electrode/electrolyte interfacial polarization loss, thus provides superior power out performance.

## 4. Conclusions

We have successfully prepared Sm^3+^ and Nd^3+^ co-doped CeO_2_ (SNDC) and pure CeO_2_ nanocrystalline electrolytes, and evaluated the chemical properties in low temperature solid oxide fuel cells. It has been found that the SNDC and CeO_2_ electrolytes exhibit the nanocrystalline microstructures with a size of below 50 nm. The coexistence of Ce^4+^ and Ce^3+^ ions is on the surface of samples, and the SNDC has higher Ce^3+^ concentration compared with the pure CeO_2_, implying higher oxygen vacancies and oxygen ionic conductivity in SNDC, which is also confirmed by Raman spectra analysis. The fuel cell based on the nanocrystalline SNDC electrolyte can achieve excellent electrochemical performance. The peak power density of SNDC cell reaches 1070.31 mW·cm^−2^ at 550 °C, which is about 33% higher than that (804.69 mW·cm^−2^) of CeO_2_. This is mainly attributed to the high ionic conductivity of SNDC caused by Co-doping. The fuel cell with the nanocrystalline SNDC electrolyte can obtain high power output performance, suggesting the promising potential applications for LT-SOFCs.

## Figures and Tables

**Figure 1 nanomaterials-11-02231-f001:**
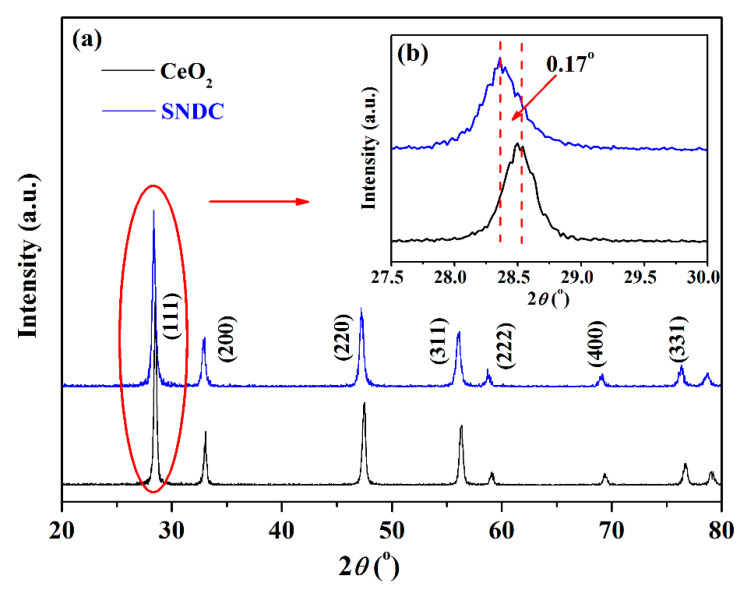
X-ray diffraction patterns of (**a**) CeO_2_ and SNDC powders and (**b**) magnified (111) crystal face of the CeO_2_ and SNDC powders.

**Figure 2 nanomaterials-11-02231-f002:**
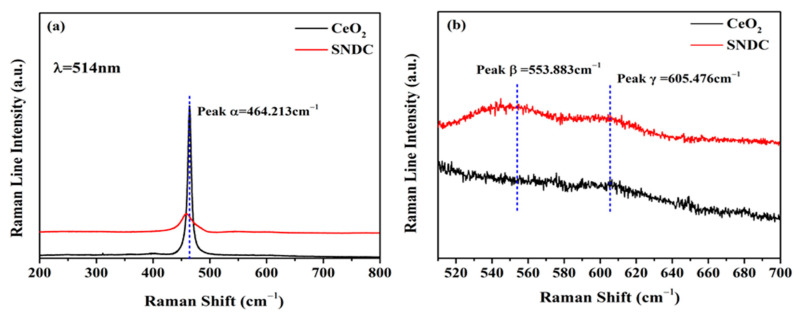
Raman spectra of (**a**) CeO_2_ and SNDC samples from 200 to 800 cm^−1^ and (**b**) magnified view with the Raman shift ranging from 510 to 700 cm^−1^.

**Figure 3 nanomaterials-11-02231-f003:**
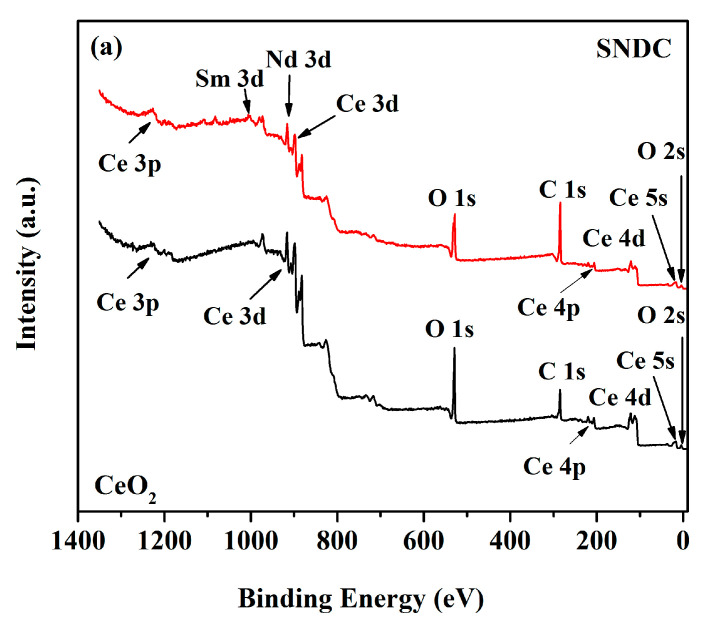
(**a**) XPS survey spectra; (**b**) O 1s core level XPS spectra and (**c**) Ce 3d core level XPS spectra of CeO_2_ and SNDC samples; (**d**) Sm 3d core level XPS spectra and (**e**) Nd 3d core level XPS spectra of SNDC sample.

**Figure 4 nanomaterials-11-02231-f004:**
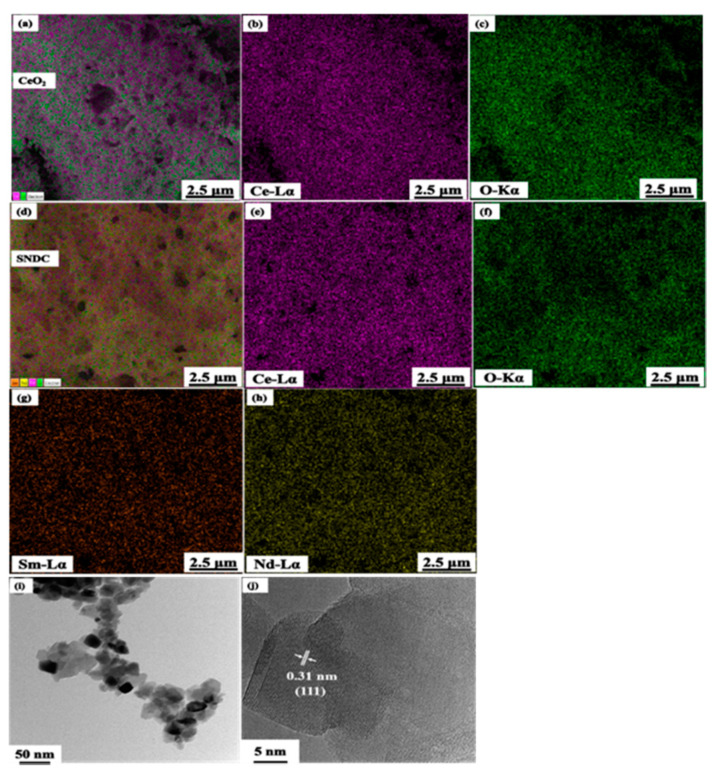
EDX element mapping of (**a**–**c**) CeO_2_ and (**d**–**h**) SNDC materials; (**i**,**j**) HR-TEM images at different magnifications of SNDC material.

**Figure 5 nanomaterials-11-02231-f005:**
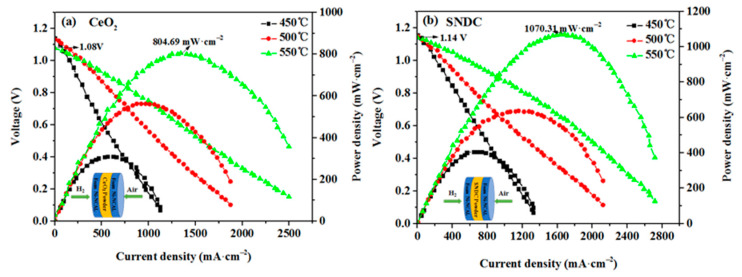
Typical I−V and I−P curves of fuel cells based on (**a**) CeO_2_ and (**b**) SNDC electrolyte operated at 450–550 °C in H_2_/air and the corresponding simple schematic diagram of the cell.

**Figure 6 nanomaterials-11-02231-f006:**
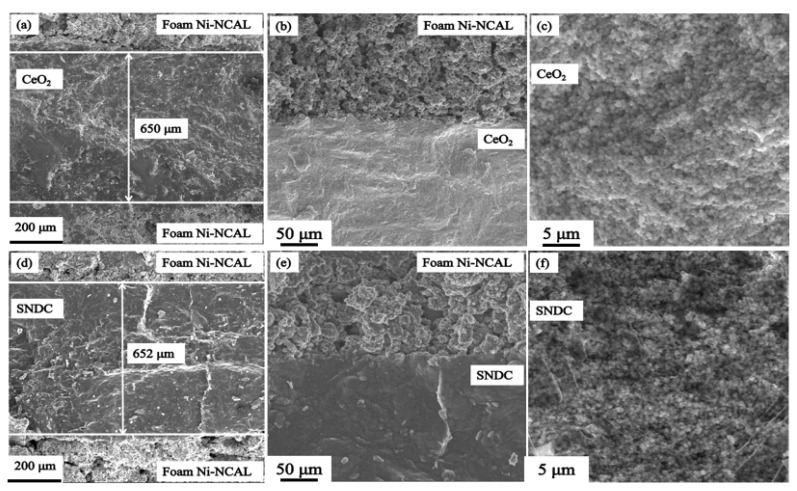
Cross-sectional SEM images of the cells with the (**a**) CeO_2_ and (**d**) SNDC electrolytes, the interface of (**b**) foam Ni-NCAL/CeO_2_ and (**e**) foam Ni-NCAL/SNDC, and the magnified (**c**) CeO_2_ and (**f**) SNDC electrolyte.

**Figure 7 nanomaterials-11-02231-f007:**
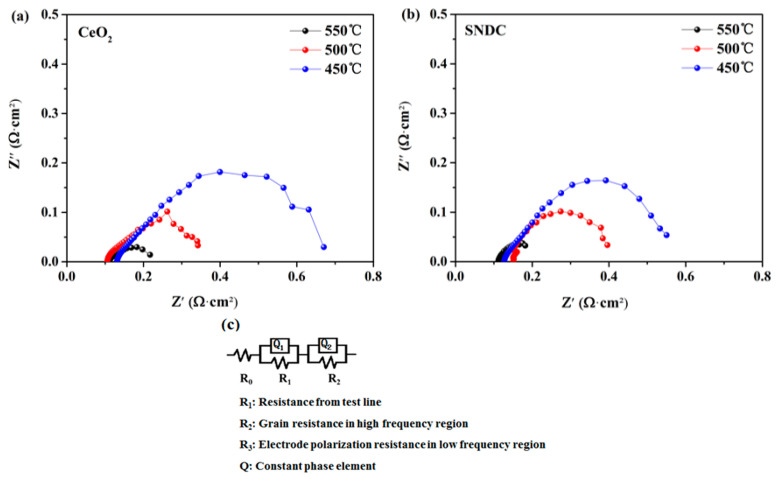
Electrochemical impedance spectra of the cells with nanocrystalline structure (**a**) CeO_2_ and (**b**) SNDC electrolytes measured at 450–550 °C; (**c**) the empirical equivalent circuit of the impedance.

## Data Availability

Data is available upon the reasonable request from the corresponding author.

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
