# Peer review of "High Performance Low-Temperature Solid Oxide Fuel Cells Based on Nanostructured Ceria-Based Electrolyte"

_nanomaterials, 2021, doi:10.3390/nano11092231_

Round 1

Reviewer 1 Report

The manuscript of Liu et al. discusses the impact of Sm and Nd-doping on the electrochemical performance of nanocrystalline ceria for SOFCs. The authors use the glycine-nitrate method to prepare Sm3+ and Nd3+ co-doped ceria powder. Detailed structural characterization confirm the successful synthesis of the material. The analysis of the fuel cell performance reveals that the co-doped samples exhibit an improvement of the performance compared to pure nanocrystalline ceria. The preparation of solid electrolytes for low temperature SOFCs is of large importance and the manuscript presents interesting results. However, there are some points, which should be addressed before the manuscript can be considered for publication. 

  1. It would be interesting to compare the results with Sm-doped ceria and Nd-doped ceria in order to elucidate the impact of the co-doping.
  2. On page 9 the authors attribute the differences of the electrochemical performance to “different electrolyte properties”. What do they mean with properties? Are the differences related to the doping, the lattice expansion, or the size of the nanocrystallites?
  3. Same on page 10, line 300: Please discuss in more detail which structural properties are responsible for the difference in R_p? Is it the grain size, the doping, or the higher ionic conductivity, or something else?
  4. In the conclusions, the author claim that the doping increases the surface oxygen exchange rate. However, a direct experimental proof is missing.
  5. There are some sentences, which are not well written. I recommend to rephrase them:
  • Page 8, line 255 “… of the cell nanocrytalline electrolytes have excellent gastight enough”
  • page 9, line 279: “…, and shows that their morphology has a packing powder particles structure when not sintered at high temperatures and the nanoparticle electrolyte particles are below 50 nm”

Author Response

Response to the Reviewer1

Comments:

This work reported properties of Sm-Nd co-doped CeO2 electrolyte fabricated by a glycine-nitrate process. In addition, Sm-Nd co-doped CeO2 was applied as an electrolyte of a low temperature solid oxide fuel cell. Results were solid and systematically carried out. However, some modifications are required.

1) It would be interesting to compare the results with Sm-doped ceria and Nd-doped ceria in order to elucidate the impact of the co-doping.

Response: Thank the reviewer very much for this important and meaningful comment. Two electrolyte materials, CeO2 and Sm-Nd co-doped CeO2, are selected for comparison because the ionic conductivity after double doping is higher than that of single doping or undoping, which is also fully proved in terms of power density. Taking Sm-doped ceria as an example, the highest power density in our test is 901.56 mW/cm2, while the maximum power density of double doping is about 1070.31 mW·cm-2. Your comments and suggestions are of great research value. In the future research, we will further study the electrochemical performance of Sm-doped and Nd-doped cerium oxide electrolyte.

2) On page 9 the authors attribute the differences of the electrochemical performance to “different electrolyte properties”. What do they mean with properties? Are the differences related to the doping, the lattice expansion, or the size of the nanocrystallites?

Response: Thank the reviewer very much for the good advices. Taking into consideration the suggestion and comment from reviewer, we have revised this part of the manuscript. The details are as follows:

“Therefore, the significant difference in electrochemical performance of the cells can be attributed to the change in ionic conductance caused by lattice expansion after double-doping.”

3) Same on page 10, line 300: Please discuss in more detail which structural properties are responsible for the difference in Rp? Is it the grain size, the doping, or the higher ionic conductivity, or something else?

Response: Thank the reviewer very much for his good advices and kind reminding. According to the reviewer’s suggestion, we have corrected this sentence in the revised manuscript. The details are as follows:

    “According to the results of the experimental data of Raman Spectra and XPS analysis, it can be seen that the difference in Rp of the two electrolytes can be attributed to the different electrochemical performance in the three-phase interface resulting from the defect spaces owing to extrinsic oxygen vacancies or structural defects caused by Sm3+ and Nd3+ dopants in ceria lattice. After double doping, the active site of the interface between electrolyte and electrode increases, and the three-phase interface increases, which eventually leads to the decrease of the Rp of the cell. In addition, as for SNDC, the ionic activity increases with the increase of test temperature, so the Rp shows a decreasing trend.

4) In the conclusions, the author claim that the doping increases the surface oxygen exchange rate. However, a direct experimental proof is missing.

Response: Thanks for raising this important point. According to the suggestions of reviewers, after repeated deliberation and thinking, we found that it was really far-fetched to add the content about ionic conductivity in the conclusion, so we revised this part in the manuscript.

5) There are some sentences, which are not well written. I recommend to rephrase them:

Page 8, line 255 “… of the cell nanocrytalline electrolytes have excellent gastight enough” page 9, line 279: “…, and shows that their morphology has a packing powder particles structure when not sintered at high temperatures and the nanoparticle electrolyte particles are below 50 nm”

Response: Thank the reviewer very much for his good advices and kind reminding. According to the reviewer’s suggestion, these two sentences have been revised in the manuscript. The details are as follows:

    (1) “that the nanoelectrolytes are effective at separating H2 and air from each side of the cell”.

    (2) “From Figure 6c, f, the pure CeO2 and SNDC electrolytes display the nanocrystalline mi-crostructures, and the nanocrystalline electrolyte particles are below 50 nm”.

Reviewer 2 Report

This work reported properties of Sm-Nd co-doped CeO2 electrolyte fabricated by a glycine-nitrate process. In addition, Sm-Nd co-doped CeO2 was applied as an electrolyte of a low temperature solid oxide fuel cell. Results were solid and systematically carried out. However, some modifications are required.

1) For understanding of readers, simple schematics of electrochemical test set up of the LT-SOFCs is required.  

2) Authors carried out XPS analysis of the Sm-Nd co-doped CeO2 electrolyte and showed spectra results. (figure 3) Also, XPS analysis can provide atomic concentration of materials. What is the atomic concentration of Sm-Nd co-doped CeO2? Does it match with the Sm0.075Nd0.075Ce0.85O2 ?

3) Authors provides results of electrochemical characterizations including OCV and polarization curves. OCVs of LT-SOFCs were measured 1.154V (450c), 1.148V (500c), 1.141V (550c), respectively. What is the theoretical OCVs at tested temperature? 

4) Also, all OCVs of LT-SOFCs with pure CeO2 electrolyte were above 1V from 450c~550c. But, pure CeO2 shows poor thermal and chemical stabilities at high temperature and reduction environments. So, CeO2 based electrolytes causes poor OCVs of SOFCs. Are there reasons for high OCVs of fuel cells with pure CeO2 electrolyte?

5) What about the long term stability of this LT-SOFCs? 

6) Equivalent circuit models for investigations in figure 7 should be added.

7) In figure 7 a), the ohmic resistance at 500 c looks smaller than the ohmic resistance at 550c. Also, figure 7 b), the ohmic resistance of the LT-SOFC at 450 c is smaller than the ohmic resistance of the LT-SOFC at 500 c. Are there some reasons of these results?

Author Response

Response to the Reviewer2

Comments:

The manuscript of Liu et al. discusses the impact of Sm and Nd-doping on the electrochemical performance of nanocrystalline ceria for SOFCs. The authors use the glycine-nitrate method to prepare Sm3+ and Nd3+ co-doped ceria powder. Detailed structural characterization confirm the successful synthesis of the material. The analysis of the fuel cell performance reveals that the co-doped samples exhibit an improvement of the performance compared to pure nanocrystalline ceria. The preparation of solid electrolytes for low temperature SOFCs is of large importance and the manuscript presents interesting results. However, there are some points, which should be addressed before the manuscript can be considered for publication. 

1) For understanding of readers, simple schematics of electrochemical test set up of the LT-SOFCs is required. 

Response: Thank the reviewer very much for his good advices. According to the reviewer’s suggestion, we have included simple schematics of electrochemical test in the manuscript. 

2) Authors carried out XPS analysis of the Sm-Nd co-doped CeO2 electrolyte and showed spectra results. (figure 3) Also, XPS analysis can provide atomic concentration of materials. What is the atomic concentration of Sm-Nd co-doped CeO2? Does it match with the Sm0.075Nd0.075Ce0.85O2?

Response: Thanks for raising this important point. The atomic concentration of samarium and neodymium in the XPS test is 2.35%, which is greater than the concentration of samarium and neodymium in the actual SNDC. Because the XPS test depth is 1-10 nm, and powder samples are used in the test process, the test will not be the same as the actual atomic concentration.

3) Authors provides results of electrochemical characterizations including OCV and polarization curves. OCVs of LT-SOFCs were measured 1.154V (450oC, 1.148V (500 oC), 1.141V (550 oC), respectively. What is the theoretical OCVs at tested temperature?

Response: Thank the reviewer very much for this important and meaningful comment. We use the Nernst equation(E=E0+ )to calculate the theoretical value of OCV, The theoretical value of OCVs are about 1.23V (450℃), 1.2V (500℃), 1.17V (550℃) at the test temperature.

4) Also, all OCVs of LT-SOFCs with pure CeO2 electrolyte were above 1V from 450 oC ~550 oC. But, pure CeO2 shows poor thermal and chemical stabilities at high temperature and reduction environments. So, CeO2 based electrolytes causes poor OCVs of SOFCs. Are there reasons for high OCVs of fuel cells with pure CeO2 electrolyte?

Response: Thank you very much for this insightful comment. Pure CeO2 electrolyte cells’ OCV is still greater than 1V, which conflicts with the traditional SOFC cells theory, but through the literature research[1-2], we found that in the new type of fuel cell system, CeO2 in contact with the anode side is reduced by H2 to form Ce3+ and release free electrons, the surface conductivity is formed, and additional electrons simultaneously lead to the n-type conductive anode side of CeO2. On the other hand, CeO2 side shows pore conduction on the air, i.e. p-type conduction, while CeO2 side is reduced to electron (n-type conduction) by H2 on the anode. In this way, a p-n junction is formed between two parts of the CeO2 electrolyte. It can promote the ion conductance, inhibit electronic conductivity, which may be one of the reasons why the open circuit voltage of CeO2 cell is higher than 1V.

[1] Wang, B.;  Zhu, B.;  Yun, S.;  Zhang, W.;  Xia, C.;  Afzal, M.;  Cai, Y.;  Liu, Y.;  Wang, Y.; Wang, H. Fast ionic conduction in semiconductor CeO2-δ electrolyte fuel cells. NPG Asia Materials 2019, 11, 51-62.

[2] Rauf, S.;  Zhu, B.;  Shah, M. A. K. Y.;  Tayyab, Z.;  Attique, S.;  Ali, N.;  Mushtaq, N.;  Asghar, M. I.;  Lund, P. D.; Yang, C. P. Low-temperature solid oxide fuel cells based on Tm-doped SrCeO2-δ semiconductor electrolytes. Materials Today Energy 2021, 20, 100661-100672.

5) What about the long term stability of this LT-SOFCs?

Response: Thank the reviewer very much for this insightful suggestion. Although we did not do long-term stability tests, we repeated IV/IP and impedance tests on the same battery, and the performance of the battery did not decrease significantly during this process. So, in the next article we will further test and study the long-term stability of LT-SOFCs.

6) Equivalent circuit models for investigations in figure 7 should be added.

Response: Thanks for your good advices. Taking into consideration the suggestion and comment from reviewer, we have added the equivalent circuit diagram in the revised manuscript. The details are as follows:

“The equivalent circuit obtained according to the impedance spectrum fitting is shown in the figure 5c. The formula used for fitting is R0(Q1R1) (Q2R2), where R0 is the overall ohmic resistance, R1 and R2 correspond to the high and low frequency resistance arcs from the electrode, respectively, and Q is a constant phase element. The high frequency (R1, Q1) arc is usually attributed to ion charge transfer process and the low frequency (R2, Q2) arc is commonly associated with oxygen dissociation and surface diffusion processes.”

7) In figure 7 a), the ohmic resistance at 500oC looks smaller than the ohmic resistance at 550 oC. Also, figure 7 b), the ohmic resistance of the LT-SOFC at 450oC is smaller than the ohmic resistance of the LT-SOFC at 500 oC. Are there some reasons of these results?

Response: Thank the reviewer very much for his good advices. Normally ohm impedance value decreased with the increase of temperature, but in our test results do not conform to the rules, the main reason may be due to the battery test is using a metal fixture, metal fixture at test temperature impedance changes have some uncertainty, finally made an impact on the cell ohmic impedance, we tested the different cells of the same material and got the similar results.

Round 2

Reviewer 2 Report

All results of the report are solid and deserve publication. I recommend that this work can be published.